# Adjustment Criteria for Air-Quality Standards by Altitude: A Scoping Review with Regulatory Overview

**DOI:** 10.3390/ijerph22071053

**Published:** 2025-06-30

**Authors:** Lenin Vladimir Rueda-Torres, Julio Warthon-Ascarza, Sergio Pacsi-Valdivia

**Affiliations:** 1Programa de Doctorado en Ingeniería y Ciencias Ambientales, Universidad Nacional Agraria La Molina, Lima 15024, Peru; sergiopacsi7@gmail.com; 2Centro de Investigación de Energía y Atmósfera, Universidad Nacional de San Antonio Abad del Cusco, Cusco 08003, Peru; julio.warthon@unsaac.edu.pe

**Keywords:** altitude, air quality standards, pollution, adjustment, scoping review

## Abstract

Air-quality standards (AQS) are key regulatory tools to protect public health by setting pollutant thresholds. However, most are based on sea-level data. High-altitude (HA) environments differ in atmospheric conditions, influencing pollutant behavior and human vulnerability. These differences have prompted proposals for altitude-specific AQS adjustments. This systematic review identifies models and criteria supporting such adaptations and examines regulatory air-quality frameworks in countries with substantial populations living at very high altitudes (VHA). This review follows PRISMA-P guidelines, focusing on studies examining AQS adjustment approaches based on altitude. The Population/Concept/Context (PCC) framework was used to define search terms: population (AQS), concept (air pollutants), and context (altitude), with equivalents. The literature was retrieved from PubMed, Scopus, Web of Science, and Gale OneFile: Environmental Studies and Policy. A total of 2974 articles were identified, with 2093 remaining after duplicate removal. Following title and abstract screening, 2081 papers were excluded, leaving 12 for full-text evaluation. Ultimately, six studies met the eligibility criteria. Three studies focused on adjustment models based on atmospheric conditions, such as temperature and pressure changes, while the other three examined human physiological responses, particularly the increased inhaled air volume. China, Peru, and Bolivia have the largest populations living above 3500 m a.s.l., yet none of these countries have specific air-quality regulations tailored to HA conditions. The review underscores the necessity for tailored AQS in HA environments, highlighting specific criteria related to both atmospheric conditions and human physiological responses.

## 1. Introduction

The World Health Organization (WHO) estimates that 99% of the global population breathes air with pollutant levels exceeding recommended limits, making it the primary environmental medium for pollutant exposure and the second leading risk factor for death worldwide [1]. Outdoor air pollution is responsible for nearly 5.2 million deaths annually, based on recent estimates [2], and is also associated with high morbidity across a wide range of adverse health outcomes beyond classical cardiorespiratory diseases, including metabolic disorders, adverse birth outcomes, and impacts on mental health and cognitive development [3]. This critical situation underscores the need to strengthen public health policies, particularly in low- and middle-income countries where the health impacts are more severe and progress in pollution reduction has been limited [4,5].

Air-quality standards (AQS) are legally binding instruments and essential components of public health strategies aimed at managing exposure to air pollutants [6]. They establish pollutant concentration thresholds and objectives to minimize the risk of adverse health effects [7,8]. Most countries develop their national (or federal) AQS primarily based on international guidelines provided by organizations such as the World Health Organization (WHO), the U.S. Environmental Protection Agency (EPA), and the European Environment Agency (EEA), among others [9,10].

These guidelines offer evidence-based recommendations derived primarily from a systematic analysis of environmental epidemiology studies conducted predominantly in metropolitan areas with adequate resources and well-established monitoring networks [10,11]. However, these conditions may not be replicable in developing countries, which often feature diverse environmental settings and limited air-quality management infrastructure [12]. In this context, the guidelines highlight the need to adapt air-quality target values to local conditions before implementing international standards [13].

High-altitude (HA) environments can significantly influence pollution levels and impact population exposure [14]. A 1978 report by the EPA highlighted that HA atmospheric conditions, such as reduced barometric pressure, lower temperatures, increased solar radiation, and other factors, affect pollutant dynamics through various mechanisms [15]. These include intensifying photochemical reactions, increasing the frequency of thermal inversions, promoting aerosol formation, dispersion, deposition, condensation, and altering vehicle emissions, among others [16,17].

The respiratory physiology of native HA populations is another critical factor to consider in this context [18,19]. Over generations, these populations have developed anatomical and physiological adaptations that enhance lung capacity, allowing them to cope with hypoxic conditions at altitudes [20]. As a result, they exhibit higher minute ventilatory rates than sea-level populations, approximately twice as much at rest [21], which leads to the inhalation of larger air volumes on a daily basis. This physiological trait may increase their cumulative exposure to airborne pollutants, even when ambient concentrations are similar [22].

These conditions underscore the numerous variables that pose a complex challenge when considering the need for altitude-specific AQS [23]. However, previous research in the biological field has demonstrated that adjusting parameters based on biological markers is feasible [24]. A notable example is the classification criteria for anemia at HA, where a non-linear increase in blood hemoglobin (Hb) levels has guided the establishment of specialized reference values and clinical decision thresholds for these populations [25].

Altitude-dependent pollution dynamics are particularly critical for rapidly expanding HA cities, where emissions and meteorological challenges are exacerbated by urbanization and population growth [26]. Approximately 81 million people live at altitudes above 2500 m above sea level (m a.s.l.), with major population centers located in low- and middle-income regions, such as the Tibetan Plateau in China and the Andean countries in South America [27].

It is well known that these countries often enforce less stringent regulations, frequently exceeding both AQS, when available, as well as WHO guidelines [28]. The combined effect of overexposure and altitude may help explain the higher prevalence of respiratory diseases associated with air pollution, such as asthma, among individuals living at HA compared to those at sea level [29,30]. Furthermore, this issue remains largely underexplored, exposing a social inequality gap driven by inadequate regulatory frameworks in these regions.

This scoping review aims to identify the available literature that addresses setting AQS that adapt to altitude environments and the factors influencing it. Additionally, it seeks to summarize existing regulations regarding AQS in major HA countries, discussing the criteria and conditions relevant to protecting the health of populations living at VHAs. This review highlights gaps and opportunities for developing effective air-quality guidelines tailored to specific contexts and for establishing public policies and regulations that prioritize health protection.

## 2. Methods

A scoping review was conducted to inform the proposed models and criteria for adjusting air-quality standards (AQS) to altitude, complemented by an overview of AQS regulations applied in countries with a higher proportion of the population living in HA cities.

### 2.1. Protocol Registration and Search Strategies

This study follows the PRISMA-P (Preferred Reporting Items for Systematic Review and Meta-Analysis Protocols) guidelines to ensure transparency, reproducibility, and quality throughout the review process. The Population/Concept/Context (PCC) framework was utilized to define and organize the search terms. These terms included population (air-quality standards), concept (air pollutants), and context (altitude), along with their corresponding equivalents identified through the PubMed MeSH and PAHO DeCS web portals. The protocol was designed a priori and registered on the Open Science Framework (OSF) platform, available at https://osf.io/c3fnh/ (accessed on 27 February 2025).

This review was conducted using the literature from key databases, including PubMed, Scopus, Web of Science, and Gale OneFile: Environmental Studies and Policy. Detailed search algorithms are provided in Appendix A. The search strategy was designed to avoid duplication of existing systematic reviews, ensuring the novelty and relevance of the present study. For the overview of AQS regulations, the study focused on countries with the largest populations living above 3500 m a.s.l.: China, Peru, and Bolivia [27]. Reports on air-quality regulations from the environmental agency repositories of these countries were analyzed and summarized alongside contextual studies.

### 2.2. Eligibility Criteria and Selection Studies

This study included all study designs limited to studies published in English up to November 2024, covering the entire period from the implementation of the first formal AQS globally to the present. The inclusion criteria were as follows:Studies proposing, modeling, or establishing criteria for altitude-adjusted AQS in outdoor air, considering any methodological approach.The primary pollutants (criteria pollutants) [31] of interest included particulate matter (PM), nitrogenous species, carbon oxides, and ozone.

### 2.3. Data Management

Relevant data were systematically and consistently collected from the selected studies. A standardized form was used to ensure uniformity in data extraction among reviewers. This form included specific fields to capture critical information, such as study characteristics (author, year of publication, country of origin), study design, study population, contaminants criteria for AQS adjustment, and primary results or outcome measures.

Initially, two reviewers independently screened the information by titles and abstracts. If a decision could not be made based on these, the full text of the article was reviewed. Any disagreements between reviewers were resolved by consensus or through the intervention of a third reviewer. The Rayyan QCRI web platform was used to manage the selection of articles, exclude duplicates, and record the reasons for exclusion, further ensuring the validity of our research.

In accordance with the Joanna Briggs Institute methodological framework for scoping reviews, no formal quality appraisal of the included studies was conducted. This approach is consistent with the objective of mapping the available evidence rather than assessing the risk of bias or methodological rigor [32].

## 3. Results

Our initial search identified a total of 2974 articles, but 2093 documents remained after the removal of duplicates. Following the titles and abstracts screening, 2081 papers were excluded, leaving 12 articles for full-text evaluation as shown in the flow chart (Figure 1). A subset of six studies met the eligibility criteria and were included in the final review. The main exclusion reason for these studies was the lack of a clear proposal for AQS adjustment.

### 3.1. Altitude Adjusting Criteria in AQS

Six studies were identified that directly or indirectly address criteria for adjusting air pollutant standards based on altitude (Table 1). These studies fall into two main categories based on the variables and rationale underlying their adjustment models. The first category considers atmospheric and environmental factors, such as barometric pressure, temperature, and air density. The second focuses on human physiological adaptations to altitude, including changes in respiratory volume, hemoglobin–CO dissociation, and other physiological responses.

A pivotal early insight by Goddard and Goddard (1979) proposed the application of altitude corrections to the California Ambient AQS [33]. Their approach involved converting pollutant concentrations adjusted for pressure and temperature variations at specific altitudes. As an example, they identified an 8% error lower in the AQS for hydrogen sulfide when comparing the standard at the time to its equivalent at an elevation of 2200 ft (670.5 m).

The proposal by Bravo and Urone (1981) highlighted altitude as a “Fundamental Parameter” for AQS adjustments, contextualized within the challenges posed by Mexico City, a congested capital city with elevation (2240 m a.s.l.) [34]. The approach to adjusting AQS was rationalized based on the physiological implications of reduced barometric pressure and air density on gas exchange at the alveolar–capillary human lungs interface at HA. Under these conditions, lower airway resistance reduces respiratory effort, allowing a greater volume of air to enter the lungs compared to individuals at sea level [35]. Based on this premise, the authors proposed an AQS adjustment factor using the ratio of respiratory rates under standard and HA conditions, as shown in Table 1.

Using previous data on resting ventilation volume, body measurements, and average air PM concentrations, they estimated that an average resident of Mexico City inhales an excess daily air volume of approximately 2664 L·m^−2^ of body surface area. This would correspond to an additional cumulative PM exposure dose of 690 µg per square meter of body surface area.

Another notable physiological approach was proposed by Collier and Goldsmith (1983), who focused on ambient AQS carbon monoxide (CO) levels at HA [36]. It is well established that CO binds to hemoglobin (Hb) with an affinity 200 to 300 times greater than that of oxygen [37], potentially leading to hypoxic effects [38]. The key assumption is that HA conditions would require stricter CO AQS due to the reduced partial pressure of oxygen (PO_2_), which further enhances CO’s affinity for Hb, resulting in higher carboxyhemoglobin (COHb) levels [39].

The CO AQS established in the U.S. set the threshold at 10 mg·m^−3^ (9 ppm) for an 8 h exposure [40]. However, the states of California and Nevada tightened it to 6 ppm (5.5 mg·m^−3^) for the touristy Lake Tahoe Basin, elevated at 1900 m a.s.l. [41]. In that context, Collier and Goldsmith evaluated the appropriateness of these adjustments with a theoretical mathematical model that integrated the relationships among O_2_, CO, and Hb through the Coburn–Forster–Kane (CFK) equation, the Roughton–Darling equation, the oxygen dissociation curve (ODC), alveolar ventilation equations, and formulas for barometric pressure and pCO as functions of altitude [36].

**Table 1 ijerph-22-01053-t001:** Characteristics of the included documents from the scoping review.

Author Details	Methodology and Pollutant Assessed	Basis for Model Adjustment	Model Equation for AQS Adjustment
Goddard and Goddard, 1979—USA [33]	Report with calculus. Exemplified with H_2_S.	Concentration conversion by change of pressure and temperature.	$Equivalent\;AQS\;\frac{g}{m^{3}}=AQS\left(\frac{moles\;of\;air}{m^{3}}\right)\left(\frac{gm\;of\;pollutant}{mole}\right)$
Bravo and Urone, 1981—Mexico [34]	Simulation study using TPS.	Hyperventilation at altitude increases risk of pollutant exposure.	$Adjusted\;AQS=Sea\;level\;AQS\left(\frac{Standard\;respiratory\;rate}{Altitude\;respiratory\;rate}\right)$
Collier and Goldsmith, 1983—USA [36]	Numerical simulation modeling of CO.	Less oxygen partial pressure at altitude increases risk of carboxyhemoglobin.	Coburn, Forster, and Kane (CFK) Equation
Madueño et al., 2020—Germany [42]	Observational field study measuring black carbon exposure.	Hyperventilation and modes of transport at altitude increase risk of pollutant exposure.	$RDD=\Sigma C_{ij}\cdot MV_{i}\cdot △t_{ij}$ where: RDD: Respiratory Deposition Dose, C: Pollutant concentration, MV: Minute ventilation, △t: Exposure time, i: Index for activity type and j: Index street segment
Bravo Alvarez et al., 2013—Mexico [43]	Simulation-based estimation using PM_10_.	General gas law. Volume concentration change at different pressures and temperatures.	$C_{L}=C_{St}\cdot \left(\frac{P_{L}}{P_{ST}}\right)\cdot \left(\frac{T_{St}}{T_{L}}\right)$ where: C: Pollutant concentration, P: Pressure, T: Temperature, L: Index for local condition and St: Index for standard condition
Yan et al., 2021—China [44]	Observational field study measuring CO and NO_2_.	Low oxygen content and ambient pressure can influence vehicle pollutant emissions.	$f=\left(\frac{E}{E_{0}}\right)$ where: f: Altitude coefficient, E: Emission factor at altitude and $E_{0}$: Base (standard) emission factor

The model’s results indicated that the 8 h and 1 h CO standards established for sea-level conditions would not require modification to ensure health protection at HA, provided the standards were expressed in volumetric terms. Although their calculations did not provide sufficient evidence to justify adjusting CO AQS in volumetric terms, they estimated that exposure to 8 ppm of CO would result in equilibrium COHb levels of 1.4% at sea level and 1.8% at 3579 m.

This contrasted with the EPA’s approach, which was based on gravimetric measurements. The rationale behind this finding was that altitude and hyperventilation proportionally decrease both inspired pCO and pO_2_, maintaining their relative balance [45]. However, it was mentioned that other factors were not considered, such as increased CO emissions due to inefficient vehicle engine combustion at altitudes, which could heighten exposure risks [46].

An indirectly adjusted measure included in this review is the study by Madueño et al. (2020), which examines black carbon (BC) exposure at HA [42]. The authors estimate the potential Respiratory Deposited Dose (RDD) of BC in individuals using different modes of transportation in La Paz and El Alto, Bolivian cities located at 3600 and 4100 m a.s.l., respectively. The study compares RDD findings with similar research conducted in low-altitude regions [47], revealing that black carbon exposure while walking in Bolivia is, on average, 2.6 times higher than at sea level.

The primary factor for calculating RDD in Table 1 is the minute ventilation rate (MV) associated with a given activity, commonly referred to as the inhalation rate [48]. MV should be considered the key variable in assessing exposure under the same air pollutant concentration [49,50]. Based on these findings, the authors highlight the urgent need for stricter and more aggressive policies and regulations on road emissions in HA cities [42].

Among the studies that adopt environmental and atmospheric approaches to altitude-based AQS adjustments, Bravo Alvarez et al. (2013) [43] is one of the most frequently cited on this topic [51,52,53]. Their approach is based on a straightforward application of the general gas law equation, proposing an adjustment factor derived from the ratio of temperature at sea level (298.15 K) to its corresponding value at a given altitude ($T_{L}$), multiplied by the ratio of atmospheric pressure at that altitude ($P_{L}$) to its standard sea-level value (1013 hPa), as shown in Table 1.

This is a simple but versatile model that can be applied to any AQS pollutant. The study exemplifies its application by adjusting the Mexican AQS for total suspended particles (TSP) and particulate matter less than 10 µm (PM_10_) across cities at different altitudes in Latin America [43]. The specific correction factors for PM_10_ range from 0.97 for Santiago de Chile (567 m a.s.l.) to 0.68 for El Alto, Bolivia (4050 m a.s.l.), indicating that as altitude increases, AQS values may be proportionally reduced. Among the Latin American cities analyzed are Mexico City (Mexico), with a factor of 0.79; Bogotá and Cali (Colombia), with 0.76; Quito (Ecuador), with 0.75; and Cusco (Peru), with 0.70 [43].

The last study included was conducted by Yan et al. (2021), focusing on air-quality HA road tunnels [44]. The current Chinese guidelines for tunnel ventilation [54] recognize the influence of altitude on exacerbated vehicle CO emissions. To address this effect, the guidelines establish a CO altitude emission coefficient, which quantifies the additional emissions produced at a given altitude compared to the same vehicle operating at sea level. According to these regulations, the CO emission coefficient under standard conditions (below 400 m a.s.l.) is set at 1, but it triples at an altitude of 4000 m [54].

Yan et al. aim to assess the validity of the existing CO altitude emission coefficient and to propose a similar coefficient for NO, which was not included in the Chinese guidelines. They measure CO and NO levels in four road tunnels at altitudes ranging from 500 to 3850 m a.s.l., considering atmospheric parameters, traffic conditions, and structural characteristics [44]. Using mass balance models, they calculate emission factors and find that the CO altitude coefficient increases nearly linearly with altitude, aligning with the values established in the guidelines within the studied altitude range. Additionally, they propose an NO emission coefficient of 0.403 g·vehicle^−1^·km^−1^ at 500 m, which increases linearly to 0.886 g·vehicle^−1^·km^−1^ at 3850 m a.s.l.

### 3.2. Altitude-Influenced Pollutants, AQS, and Regulation in Major HA Countries

The implementation of AQS levels varies significantly across countries, with differing levels of stringency in pollutant thresholds, leading to a wide range of regulations for the same pollutant [55]. This variation has been attributed to multiple factors and their interrelations, including government systems, a country’s level of development, sociolegal cultures, technical expertise, historical patterns of air-quality legislation, and other considerations [7]. However, specific conditions such as topography and altitude are often either overlooked or insufficiently addressed.

The WHO AQS (2000) guidelines acknowledge altitude as a factor influencing air pollutant concentrations; however, they do not provide specific recommendations for addressing altitude-related AQS adjustments considerations [56]. In contrast, the updated WHO 2021 guidelines do not mention altitude in this context. Despite this omission, growing evidence with strong theoretical plausibility supports the impact of altitude on pollutant concentrations, particularly for ozone (O_3_), nitrogen oxides (NO_x_), and PM [57,58,59]. These criteria pollutants are of major public concern, with particulate matter less than 2.5 µm (PM_2.5_) being the leading contributor to global morbidity and mortality [60,61], and O_3_ ranking as the second most detrimental pollutant to human health [62].

The magnitude of effects and outcomes is determined by the absolute altitude. According to the EPA, at elevations above 1500 m, significant atmospheric changes begin, with pressure decreasing by approximately 15% compared to sea level [15]. HA is defined as elevations between 2500 and 3500 m a.s.l. [63], while VHA (>3500 m) presents extreme conditions that exacerbate the impacts of various risk factors [64].

Global estimates and distributions of altitude populations have shown considerable variability over time. Earlier approaches primarily relied on inconsistent population records and incomplete country-level data [65]. In contrast, recent studies have integrated georeferenced population estimates with elevation datasets to generate refined global and national-level assessments of human populations residing at various altitudes [27]. According to these estimates, approximately 500 million people live above 1500 m, 81 million above 2500 m, and 14 million above 3500 m. The latter group is represented in Figure 2, which illustrates their distribution using a log_10_ scale to enhance visual contrast and highlight inter-country differences. Nearly 90% of this population resides in just three countries: China, Peru, and Bolivia, in that order.

In this context, to evaluate current air-quality legislation and specific regulatory approaches in countries where a significant proportion of the population resides above 3500 m a.s.l., the national AQS frameworks were examined from China, Peru, and Bolivia, focusing on their regulatory thresholds for PM, O_3_, and NO_2_ and comparing them to the WHO guide (Table 2).

China has the largest HA population globally, with 5.1 million people residing above 3500 m a.s.l., primarily due to the inclusion of the Tibetan Plateau, the highest landmass on Earth [70]. However, this represents only a very small fraction—0.37%—of the country’s total population [27]. China’s current national AQS (GB 3095-2012) [69] were issued by the Ministry of Environmental Protection in 2012 for primary pollutants and came into effect nationwide in January 2016 [71]. Since then, air quality in China has steadily improved due to the implementation of various national policies prioritizing clean air targets [72].

A distinctive feature of China’s AQS is its dual-level system, which establishes two classes of limit values for each pollutant. Class I standards apply to areas requiring special protection, such as scenic spots and nature reserves, and Class II standards apply to all other areas, including residential, mixed-use, industrial, and rural zones. The annual mean PM_2.5_ concentration limit across mainland China is set at 35 µg·m^−3^. Compliance with this standard has been relatively high; in 2022, only 18% of cities (primarily in the north, central regions, and the Sichuan Basin) recorded annual mean PM_2.5_ concentrations exceeding the national ambient AQS [73].

Despite these efforts, China’s AQS for PM pollutants remain considerably less stringent, being up to seven times more permissive than the WHO air-quality guidelines [13]. Additionally, the 8 h ozone AQS level (Class II) is the most lenient among the countries analyzed in this review. According to national regulations, provincial governments in China may establish local AQS for parameters not covered by national standards and impose stricter limits on those already regulated. For instance, Song et al. proposed adjustments for Hainan Province [74], and administrative division adjustments (ADA) have been suggested to modify environmental governance policies for local governments [52]. However, there is limited literature on the development of local AQS for HA cities in China [71].

The spatiotemporal distribution of air pollution indicates low pollution levels in the peripheral areas of HA zones [58], likely due to the sparse population and minimal industrial activity in these regions. However, seasonal pollution spikes can occur, as observed in Lhasa, one of the highest cities in China (3650 m). While the annual average PM_2.5_ concentration remains within the Chinese standard of 35 µg·m^−3^ [75], winter pollution is severe. This is primarily attributed to the widespread use of butter lamps, which significantly increase PM_2.5_ levels, raising the city’s average winter concentration to 118 ± 60 µg·m^−3^ [76].

Peru ranks second, with an estimated 4 million people living above 3500 m a.s.l., although this figure is likely underestimated due to a lack of comprehensive data [27]. The HA Peruvian population is primarily distributed along the Andean Cordillera in western South America. After Chile, Peru has the largest proportion of its territory covered by the Andes [77], a region rich in mineral resources that has driven large-scale displacement of households and communities [78,79]. Notably, the highest altitude city in the world is the Peruvian mining settlement of La Rinconada, located between 5100 and 5300 m, where gold mining activities are predominant [80]. Another notable city is La Oroya (3745 m), which has been ranked among the ten most polluted places on Earth. An international court has even assessed the state’s responsibility for the severe air contamination affecting its population [81,82].

Peru’s AQS, established in 2017, was developed following the WHO Interim Target I (2005) for PM pollutants [83]. According to air-quality monitoring data from Peru, the PM_10_/PM_2.5_ ratio for a 24h period ranges between 0.32 and 0.65, consistent with the suggested value of 0.5 for developing countries [20]. In the case of NO_2_, the annual AQS is higher than in other countries and exceeds the 2021 WHO recommendation by a factor of 10 (Table 2). Currently, there is no specific AQS regulation for HA cities. However, a legislative proposal introduced in 2016 (Draft Law 756/2016-CR) suggested an adjustment model for AQS regarding PM, NO_2_, O_3_, and SO_2_, incorporating correction factors based on air density and atmospheric pressure at each measurement altitude [84]. As of this review, no further discussion on its implementation has taken place.

Bolivia has the highest proportion of its population living above 3500 m, approximately 32.6% of its total inhabitants [27]. Additionally, it has the highest capital city in the world (La Paz, 3869 m average elevation). As is characteristic of metropolitan areas, vehicular emissions represent an important contributor to its air pollution [85]. Cochabamba (2.558 m), another Bolivian city, was listed by WHO as one of the five most contaminated cities in Latin America [86]. This is due to its complex “bowl” topography, which influences the accumulation of air pollution associated with transport emissions [87].

Bolivian AQS regulation presents a unique case, operating under two frameworks: the official 1992 General Environmental Law (N° 1333) and the Bolivian technical standards (*Normas Bolivianas*, NB) issued by IBNORCA (Bolivian Institute of Standardization and Quality) [66,67,68]. The latter is indicated only as a reference guide, without mandatory enforcement. Notably, limits for PM_2.5_ are absent in the General Law, and the PM_10_ and PM_2.5_ limits set by IBNORCA are among the most stringent compared to other countries and are remarkably close to the WHO air-quality guidelines (Table 2). The specific reasons for this significant difference remain unclear, and it is uncertain whether they are related to the altitude conditions. In contrast, the General Law establishes the most lenient AQS, with the highest limits for PM_10_, O_3_, and NO_2_ (1 h standard).

## 4. Discussion

This review assesses the available scientific evidence on the criteria and models proposed for adjusting AQS in response to altitude conditions, alongside an analysis of air-quality regulations and policies implemented in countries with significant populations residing above 3500 m a.s.l.

Globally, environmental and epidemiological studies evaluating health risks associated with pollutant exposure in HA urban populations remain scarce [88,89]. Air-quality monitoring and regulatory efforts have traditionally focused on major metropolitan areas, predominantly located at or near sea level [90]. Consequently, recommendations, regulations, and international guidelines are largely derived from studies conducted in these settings [13].

This knowledge gap may stem, at least in part, from the longstanding perception that HA environments inherently offer fresher and cleaner air due to lower population density and limited industrial activity [91]. However, several HA urban centers have undergone significant development, disrupting environmental equilibrium through increasing emissions that pose substantial public health challenges [92,93].

Furthermore, these challenges may be exacerbated by climate change, as HA environments are on the frontlines of its impacts [94]. Projected alterations in meteorological conditions, particularly temperature, wind patterns, seasonal variations, and humidity, can significantly impact the formation, dispersion, and deposition of air pollutants in HA regions [95,96,97]. Additionally, extreme heat waves, glacial retreat, and extreme weather events threaten water availability and agricultural productivity [98]. These conditions may drive increasingly complex migration dynamics, including both rural-to-urban and urban-to-rural movements, as populations seek refuge from urban expansion or pursue emerging economic opportunities [99,100]. The effects of these processes are highly region-specific and may result in either improvements or deteriorations in air quality, depending on local topography and atmospheric dynamics.

### 4.1. Altitude-Specific AQS Adjustment Proposals

This review identifies six proposals for adjusting AQS, three of which primarily focus on atmospheric and environmental external conditions, while the remaining three emphasize human physiological responses.

Within the first group, both Goddard and Bravo Alvarez et al. proposed a similar approach based on the reasoning that AQS expressed as pollutant concentrations under standard reference conditions (25 °C and 760 mmHg at sea level) should inherently yield lower equivalent concentrations as altitude increases. A comparable concept was explored by Warthon Ascarza (2023) in their doctoral thesis [101], which developed an empirical exponential function based on air density to adjust Peruvian PM AQS in its HA cities. Their adjustment factor estimated for Cusco city (0.67) closely aligns with that reported by Bravo (0.70) for the same city, providing empirical support for this method.

Although simple and, in theory, consistent, the validity of this approach would be subject to debate, as it relies on a reductionist assumption of ideal gas laws behavior [102], which may oversimplify the complex interactions present in real-world atmospheric conditions. Furthermore, the temperature, as a variable included in the equations, can fluctuate significantly throughout the day and across monitoring stations, potentially resulting in a wide range of adjustment factors for the same altitude location.

A context-specific study included in this review is that of Yan et al., which focused on the indoor air quality in HA road tunnels, a critical factor in the design and operation of tunnel ventilation systems in China [54]. Numerous studies have reported that tailpipe emissions tend to increase at higher altitudes due to the diminished performance of internal combustion engines [103,104]. This is primarily attributed to reduced oxygen availability [105], increased driving on steep gradients, greater use of brakes, and the widespread use of engines calibrated for sea-level conditions [106].

Pollutant concentrations reported in sea-level tunnels have reached extremely high levels, with PM concentrations up to 1490 µg/m^3^ and NO_2_ levels of 4982 µg/m^3^ [107]. However, evidence remains limited regarding the health effects of very short but frequent repeated exposures, or occupational exposure to such pollutant levels [108], conditions that may be further exacerbated in HA tunnels. Some jurisdictions adopt a variable AQS for tunnels, often based on or adapted from the PIARC (Permanent International Association of Road Congresses) guidelines, which typically use CO as the primary indicator of motor vehicle emissions [109]. These standards do not include specific adjustments for altitude.

The focus then shifts to human physiological responses to altitude for AQS adjustments. These approaches primarily incorporate respiratory variables to account for differences in the average volume of air inhaled, which serves as a proxy for pollutant intake. Evidence supporting these proposals has been explored. In vitro respiratory tract models have demonstrated that increasing altitude significantly enhances the deposition of inhaled particles in the respiratory system [22], making it more vulnerable and less capable of effectively expelling harmful substances [110].

The natural physiological response to altitude, known as the hypoxic ventilatory response (HVR), triggers an increase in breathing to enhance alveolar ventilation by up to fivefold in newcomers to HA environments [111]. However, distinct respiratory patterns are observed among native high-altitude populations. Tibetans exhibit high resting ventilation and a strong HVR, similar to acclimatized lowlanders, whereas Andeans tend to have lower resting ventilation and a blunted HVR [112]. In that sense, different populations should be characterized considering ethnicity, age, sex, and activity levels to estimate specific respiratory parameters [113,114,115].

Several approaches have been developed to assess the dose of air pollutant exposure [116]. Unlike the method proposed by Bravo and Urone, which relies on average daily air intake per unit of body surface area, current risk assessment guidelines recommend using the pollutant concentration in air (e.g., mg·m^−3^) as the primary exposure metric, rather than estimating dose based on inhalation rate and body weight [117]. This shift is grounded in the recognition that, particularly in adults, lung volume and respiratory function are more closely associated with height than with body mass [118]. Accordingly, approaches like that of Bravo and Urone may lead to biased exposure estimates, especially in individuals with a high body mass, because that excess weight does not contribute meaningfully to pulmonary function.

This limitation becomes especially relevant in HA populations where increased lung volumes and ventilation rates are not proportionally reflected by body weight due to reductions of fat mass, muscle mass, and body water [119,120].

In contrast, Madueño et al. employed the RDD, a refined in situ measure technique that offers a more comprehensive assessment of the interaction between air pollution and human physiology. This method, while simple, is widely accepted for estimating the potential deposition dose of inhaled pollutants along the respiratory tract and enables meaningful comparisons across different studies [121]. MV is the key physiological parameter used to calculate RDD, and its accurate estimation becomes particularly critical in HA contexts. Various studies have reported MV increases ranging from approximately 11.5% to 58%, depending on altitude level, acclimatization status, and physical activity conditions [122,123]. In these cases, the ratio or difference between observed measurements in both representative populations would provide a feasible factor for adjusting AQS.

Finally, Collier and Goldsmith highlighted the relationship between altitude and CO-Hb binding. CO toxicity primarily causes damage through anoxic injury and lipid peroxidation [124]. At high altitudes, this condition can potentially be fatal, as it clinically mimics acute mountain sickness (AMS). Several reports have documented CO-related fatalities in tents at high altitudes due to poorly ventilated spaces and low-flame camping stoves [125]. However, such cases appear to be underreported due to the overlap between CO toxicity symptoms and AMS, as well as the lack of definitive diagnostic testing [126].

Current air-quality regulations in the Lake Tahoe Basin, Nevada (1900 m a.s.l.), enforce stricter thresholds compared to the U.S. National Ambient Air Quality Standards (NAAQS) [41]. Notably, this is the only policy explicitly based on altitude conditions that was identified. In contrast, Denver, CO (1600 m a.s.l.) adheres to the CO NAAQS of 9 ppm, a limit that is frequently exceeded during the winter months [127].

### 4.2. Regulatory AQS Frameworks

In the pursuit of more context-sensitive AQS, it is crucial to focus on countries where a significant proportion of the population resides at very high altitudes (VHAs). Globally, the most concentrated high-altitude regions are the Qinghai–Tibet Plateau in Asia and the Andes in South America. Together, these two regions constitute the main VHA population hotspots, accounting for over 90% of the world’s inhabitants [27]. This geographic clustering underscores a scientific and regulatory challenge rooted in the disproportionate distribution of altitude-related environmental exposures and highlights a broader issue of research and policy inequity affecting these populations.

As illustrated by the three countries examined, their national AQS tend to align with intermediate benchmarks, generally corresponding to Interim Target-1 levels of the WHO guidelines [13]. However, most of these regulatory frameworks are relatively outdated, with an average age of over a decade. For instance, Bolivia’s air-quality legislation dates back to 1995 and notably lacks any standard for PM_2.5_, despite the country’s inherently HA geography and increasing concerns over urban air pollution. Similarly, in other VHA countries, although PM_10_, NO_2_, and CO are regulated, no observed frameworks explicitly integrate altitude or topographic factors into the definition or adjustment of ambient AQS.

One technical and logistical barrier to implementing adaptive air-quality regulation in VHA contexts is the challenge of deploying and maintaining air-quality monitoring networks in complex terrain [128]. Establishing and operating reference-grade stations at high elevations involves considerable costs, logistical constraints, and the need for altitude-corrected calibration procedures. These factors require specific attention in both standard setting and data interpretation.

In the absence of tailored regulatory responses, communities residing in VHA urban areas remain insufficiently protected under generalized AQS frameworks. This situation raises critical concerns from the perspective of environmental justice [129].

### 4.3. Implications and Research Needs

AQS serves as the core strategic framework for ambient air-quality management worldwide, forms the foundation of environmental protection efforts, and is a key regulatory tool. The complex interplay between atmospheric physics and human biology at HA underscores the need to discuss the potential of AQS adjustment for effective public health protection and air-quality monitoring.

Models that adjust AQS based solely on environmental variables, such as atmospheric pressure, require further refinement and should be complemented by physiological considerations. Since AQS are fundamentally designed to protect human health, incorporating health outcomes and physiological adaptations offers a more relevant and comprehensive basis for assessing long-term exposure risks. This perspective is consistent with the principles outlined in international air-quality guidelines [13].

The underlying rationale is that, even if lower thresholds are established for high-altitude regions through pollutant concentration conversion calculations, not exceeding these adjusted limits does not necessarily guarantee protection for local populations. This is because such thresholds may not adequately reflect either the increased physiological vulnerability or the accumulative or altered pollutant deposition patterns observed at HA. Therefore, regulatory frameworks must go beyond environmental metrics and integrate human health data to ensure a truly protective standard.

Additionally, emerging atmospheric phenomena are further revealing the implications of HA conditions. For instance, recent studies on new particle formation (NPF) have shown that nucleation events occur more frequently at HA [130,131], particularly in the Southern Hemisphere [132]. PF represents a major source of atmospheric ultrafine particles (<100 nm), accounting for up to 50% of aerosol number concentrations in the troposphere [133]. These findings underscore the importance of conducting assessments at regional scales and emphasize the promise of advanced tools, such as low-cost sensors and machine-learning algorithms, for capturing, modeling, and predicting the complex interplay among altitude, atmospheric chemistry, and human health [134,135].

## 5. Strengths and Limitations

This study contributes to the growing body of literature addressing altitude-specific considerations for AQS. One of its key strengths lies in its integrative approach, which combines environmental, physiological, and regulatory perspectives to underscore the need for differentiated AQS in HA environments. The comparative analysis across countries further enhances the relevance and applicability of the findings.

However, several limitations must be acknowledged. Despite a comprehensive and sensitive search strategy, there was a limited number of studies that met the inclusion criteria for this review. Among the few articles identified, many differ considerably in their methodological approaches, and half were published in the 1980s. These factors reflect a lack of recent evidence, limit the possibility of conducting direct comparisons, and reduce the overall generalizability of the conclusions.

Additionally, the regulatory analysis focused exclusively on the three most populated countries with large populations living at VHA, due to their specific relevance. However, other countries with significant populations residing at high and mid-altitudes, such as Colombia, Mexico, and Ethiopia, were not included. These contexts and their regulatory frameworks could provide valuable insights. Future studies are encouraged to address these gaps and expand the geographical scope of analysis.

## 6. Conclusions

This scoping review provides the first comprehensive assessment of criteria and models proposed for adjusting AQS in altitude environments, systematically identifying and categorizing these approaches. By expanding existing knowledge, this research offers a valuable framework for future studies aimed at developing robust, evidence-based AQS tailored to the unique challenges of HA regions.

The complex interplay among atmospheric, environmental, and physiological factors highlights the need for continued discussion on AQS adjustments to strengthen public health protection and monitoring. While air-quality policies have contributed to pollution reduction in some areas, achieving truly safe AQS remains a significant global challenge.

## Figures and Tables

**Figure 1 ijerph-22-01053-f001:**
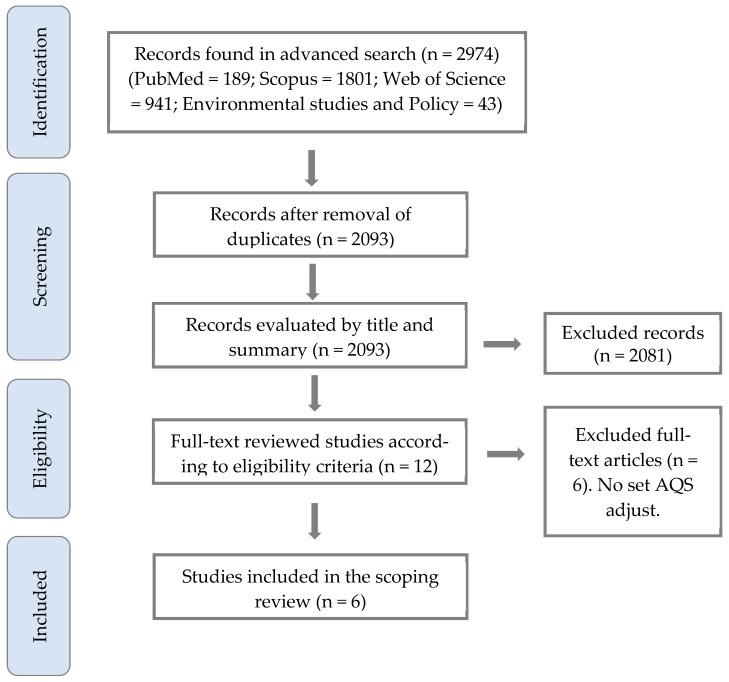
PRISMA flowchart for study selection and screening.

**Figure 2 ijerph-22-01053-f002:**
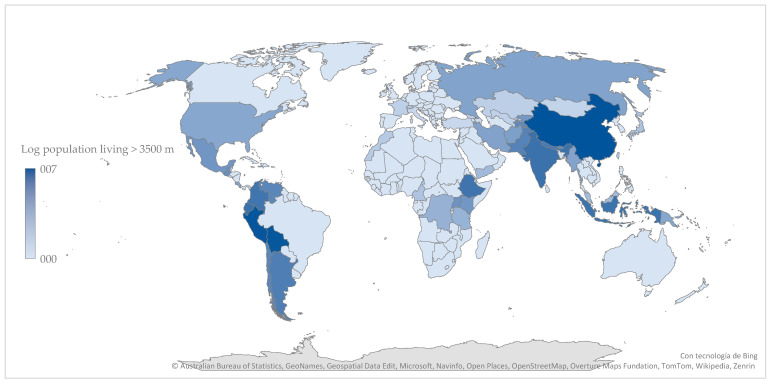
Global distribution of population living at altitudes ≥3500 m (log_10_ scale). The three countries with the largest populations at this altitude are China, Peru, and Bolivia. Data sourced from Tremblay and Ainslie [27].

**Table 2 ijerph-22-01053-t002:** Comparative AQS for key HA pollutants in countries with large populations living above 3500 m a.s.l.

Contaminant	Average Time	Bolivia		Peru	China	WHO Guide 2021
		Permissible Air-Quality Limits ^1^	Maximum Limits for Atmospheric Pollutants ^2^	Air-Quality Standards ^3^	Ambient Air-Quality Standards ^4^	
PM_2.5_, µg·m^−3^	Annual	--	10	25	Class I: 15	5
					Class II: 35	
	24 h	--	25	50	Class I: 35	15
					Class II: 75	
PM_10_, µg·m^−3^	Annual	50	20	50	Class I: 40	15
					Class II: 70	
	24 h	150	50	100	Class I: 50	45
					Class II: 150	
TPS, µg·m^−3^	Annual	75	75	--	--	--
	24 h	260	--	--	--	--
O_3_, µg·m^−3^	1 h	236	--	--	Class I: 160	--
					Class II: 200	
	8 h	--	100	100	Class I: 100	100
					Class II: 160	
	Annual	--	60	--	--	--
NO_2_, µg·m^−3^	1 h	400	200	200	200	--
	24 h	150	150	--	80	25
	Annual	---	40	100	40	10

^1^ Regulations on Air Pollution. Environmental Law No. 1333. ^2^ Bolivian Standard NB 62011:2018 [66], NB 62014:2018 [67], and 62018:2018 of IBNORCA [68]. ^3^ Ministerial Resolution No. 94-2017-MINAM. ^4^ GB 3095-2012 standard [69].

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
