# Peer review of "Adjustment Criteria for Air-Quality Standards by Altitude: A Scoping Review with Regulatory Overview"

_ijerph, 2025, doi:10.3390/ijerph22071053_

Round 1

Reviewer 1 Report

Comments and Suggestions for Authors

This systematic review identifies models and criteria supporting such adaptations and examines regulatory air quality frameworks in countries with substantial populations living in very high-altitude.

First of all, the citation format is incorrect. Please follow the journal guidelines and fix similar issues throughout the text.

Also, some of the citations are missing from the text. Please fix similar issues throughout the text.

Another issue is the tone of this work is incorrect, with the use of first person such as "we". A third-person tone should be used in a scientific article.

Figure 1 shows the Global distribution of population living at altitudes ≥3,500 m (log₁₀ scale). Please include the direction and scale into this map for better descriptions.

The discussion section is not written in a clear and easy to understand way. Please consider to rewrite the entire section and include the measures taken by different countries in a sub section.

The author did not include the limitations of this study in the conclusion section, which is necessary to reflect in this work.

Author Response

We sincerely thank the revision and the editorial team for their time, thoughtful evaluation, and constructive feedback on our manuscript. We truly appreciate the opportunity to improve our work based on your valuable insights. We have carefully addressed all comments raised by the reviewers. As part of our resubmission, we are providing the new versions of the revised manuscript, with changes (ijerph-3649864 V2)

Below, we provide a detailed, point-by-point response to each comment, indicating how and where the manuscript has been revised accordingly.

Comment 1. First of all, the citation format is incorrect. Please follow the journal guidelines and fix similar issues throughout the text.
Response:
Thank you for pointing this out. We have carefully revised the manuscript to ensure that all in-text citations and references strictly adhere to the journal’s citation guidelines. The MDPI Reference List and Citation Guide were used as a reference throughout the revision process.

Comment 2.  Also, some of the citations are missing from the text. Please fix similar issues throughout the text.
Response:
Thank you for your observation. We conducted a thorough review of the manuscript to ensure that all cited references are properly included and consistently matched in the text. Any missing or inconsistent citations have been corrected accordingly.

Comment 3. Another issue is the tone of this work is incorrect, with the use of first person such as "we". A third-person tone should be used in a scientific article.
Response:
Thank you for the suggestion. We have revised the manuscript to eliminate the use of first-person expressions and to ensure that the tone aligns with formal scientific writing conventions. Third-person phrasing is now used consistently throughout the text.

Comment 4. Figure 1 shows the Global distribution of population living at altitudes ≥3,500 m (log₁₀ scale). Please include the direction and scale into this map for better descriptions.
Response:
Thank you for your observation. We have revised Figure 1 to include both a north arrow and a scale bar, improving the geographic interpretability and descriptive clarity of the map.

Comment 5. The discussion section is not written in a clear and easy to understand way. Please consider to rewrite the entire section and include the measures taken by different countries in a sub section.
Response:
We appreciate the reviewer’s observation regarding the clarity and structure of the Discussion section. In response to this comment, as well as similar feedback raised by all four reviewers, we have fully reorganized the Discussion to improve its coherence and readability. The revised section is now structured into four subsections: (1) Altitude-Specific AQS Adjustment Proposals, (2) regulatory frameworks, (3) Implications and Research Needs, and (4) strengths and limitations. This format allows for clearer presentation and includes a dedicated subsection on measures taken by different countries, as recommended.

Comment 6. The author did not include the limitations of this study in the conclusion section, which is necessary to reflect in this work.
Response:
Thank you for the helpful observation. In response, we have added a new subsection titled “Strengths and Limitations” to appropriately address the boundaries of the study. While our findings offer novel insights, we acknowledge certain limitations, such as the limited number of available studies on this topic. This addition contributes to a more balanced and transparent interpretation of the results.

Reviewer 2 Report

Comments and Suggestions for Authors

First, I would like to begin by stating that the subject is of considerable relevance, not only for high-altitude regions in low- and middle-income countries but also for all countries with elevated locations. The discussion regarding stricter air quality thresholds at higher altitudes is not only important but absolutely necessary, and often overlooked. Overall, I congratulate the authors on this work; the manuscript is clearly written and provides useful and important information, despite the limited number of studies included in the review. However, I recommend major revisions due to several points that I believe need to be more thoroughly addressed. I also felt that the discussion section largely repeated the results, making it evident that this work would benefit from being presented as a single “Results and Discussion” section, avoiding unnecessary repetition. Below are my suggestions.

In the abstract (line 20), you refer to the database as “MEDLINE,” while in the Supplementary Material and methodology section it appears as “PubMed.” Please standardize the terminology throughout the manuscript.

In lines 37–39, the claim that outdoor air pollution causes “nearly 7 million excess deaths annually” is incorrect. According to the Health Effects Institute (2024), a reference you cited early in the introduction, the State of Global Air report actually estimates 8.1 million total air pollution-related deaths, of which approximately 64% are due to ambient (outdoor) pollution. Please revise this information accordingly.

In lines 75–77, the manuscript states that populations at high altitudes inhale larger volumes of air, but this claim lacks specific numbers. Please add quantitative support here. For example, how much greater is the volume of air inhaled in m³/day or per minute?

In line 95, the authors state: “It is well known that these countries often enforce less stringent regulations, frequently exceeding both AQS and WHO guidelines.” Please make it clear to the reader that they are exceeding their own AQS, since air quality standards are determined individually by each country—when such standards even exist.

In the begining of the introduction, air pollution is discussed only in terms of mortality, but for the relevance of this study, I believe it is important for the authors to highlight that air pollution has been associated with numerous health outcomes, going well beyond the classical cardiorespiratory outcomes and mortality alone (some relevant references to understand this context are 10.1016/j.envres.2021.111487; 10.1016/j.apr.2024.102377).

In lines 142–143, you state that the review focused on particulate matter, nitrogenous species, carbon oxides, and ozone. However, one included study (Goddard & Goddard, 1979) deals with H₂S, a sulfur compound. This inconsistency suggests a need to revise the eligibility criteria to include sulfurous species or to justify this inclusion explicitly. I suggest that you include sulfurous species, both to account for the Goddard study and to reflect the six main criteria air pollutants when discussing air pollution and health impacts.

Your methodology section does not mention any quality appraisal of the included studies. While scoping reviews often exclude formal quality assessments, this omission should be acknowledged and justified.

Lines 161, 169, 193, among others, show “Error! Reference source not found.” Please revise the manuscript and correct these formatting issues before resubmission.

Some tables are poorly formatted—for example, Table 2—and important elements such as unit consistency, alignment, and column headings need adjustment. In addition, the caption for Table 2 refers to “particulate matter AQS,” but the table addresses all pollutants, not only particulate matter. Please revise this accordingly.

In Table 1, the phrase “rationale of proposed” is poorly worded in English and should be revised.

The paragraph between lines 219–228 is repeated. The previous paragraph is exactly the same. Please correct this duplication.

In line 299, the authors state: “The WHO AQS (2000) guidelines acknowledge altitude as a factor influencing air pollutant concentrations.” What about the more recent WHO AQS? The 2021 guidelines make no mention of altitude in this context—this should be clarified.

In lines 315–318, the description of Figure 1, which concerns the population living at ≥3,500 m, lacks sufficient detail in the main text.

In lines 348–349, the authors write: “China’s air quality standards (AQS) for particulate matter pollutants remain less stringent than the WHO air quality guidelines.” They are not only less stringent—they are seven times more permissive, at least in the case of PM2.5. Please make this clear.

In lines 382–383, the authors state: “Peru's AQS, established in 2017, align with the WHO Interim Target I (2005) for PM pollutants.” Why are you comparing with the 2005 guidelines and not with the updated WHO AQS of 2021? Please revise this point.

In line 387–388, you mention “the WHO recommendation by a factor of ten.” Which WHO recommendation are you referring to? The outdated 2005 version or the current 2021 version? Clarify this point for the reader.

A major issue that requires attention is the high redundancy between the Results and Discussion sections. Much of the discussion simply repeats content from the results, particularly descriptions of individual studies. I recommend merging these into a single section to streamline the content and reduce repetition. In this new combined section, I also recommend adding a subsection with recommendations. These recommendations would benefit from the authors’ personal perspectives. Based on your findings, do you suggest that AQS should vary by altitude tiers (e.g., 2,000–3,500 m, >3,500 m), or should they be recalculated for each city specifically? How could such recommendations be implemented in countries with highly diverse topography? Offering your perspective here would strengthen the policy relevance of your study. Consider this, as well as any other recommendations relevant to the topic.

In lines 432–434, the authors write: “Furthermore, these challenges may be exacerbated by climate change and urban expansion, as HA environments could become increasingly sought after as refuges from extreme heatwaves.” I believe it is important not only to mention this climate migration dynamic, but also to discuss how climate change-related meteorological shifts can influence local air pollution levels. Changes in weather patterns can affect the formation, dispersion and deposition of pollutants and worsen air quality, and this behavior is region-specific (relevant references: 10.1080/09603123.2025.2486598; 10.3390/atmos16040363).

Author Response

We sincerely thank the revision and the editorial team for their time, thoughtful evaluation, and constructive feedback on our manuscript. We truly appreciate the opportunity to improve our work based on your valuable insights. We have carefully addressed all comments raised by the reviewers. As part of our resubmission, we are providing the new version of the manuscript. Below, we provide a detailed, point-by-point response to each comment, indicating how and where the manuscript has been revised accordingly.

Comment 1. In the abstract (line 20), you refer to the database as “MEDLINE,” while in the Supplementary Material and methodology section it appears as “PubMed.” Please standardize the terminology throughout the manuscript.
Response:
Thank you for this observation. We confirm that the search was conducted directly through the PubMed platform. Accordingly, we have revised the manuscript and supplementary materials to consistently refer to the database as “PubMed.”

Comment 2. In lines 37–39, the claim that outdoor air pollution causes “nearly 7 million excess deaths annually” is incorrect. According to the Health Effects Institute (2024), a reference you cited early in the introduction, the State of Global Air report actually estimates 8.1 million total air pollution-related deaths, of which approximately 64% are due to ambient (outdoor) pollution. Please revise this information accordingly.
Response:
Thank you for this important clarification. We have revised the statement to accurately reflect the data presented in the State of Global Air 2024 report. While the report estimates a total of 8.1 million deaths attributable to all sources of air pollution, approximately 64% of these are due to ambient (outdoor) air pollution. This corresponds to an estimated 5.2 million deaths annually (8.1 million × 0.64 ≈ 5.18 million).
This estimate is also consistent with earlier WHO data reporting 4.2 million deaths annually due to exposure to ambient air pollution (https://www.ccacoalition.org/news/world-health-organization-releases-new-global-air-pollution-data).
The manuscript now reads, “Outdoor air pollution is responsible for nearly 5.2 million deaths annually based on recent estimates.”

Comment 3. In lines 75–77, the manuscript states that populations at high altitudes inhale larger volumes of air, but this claim lacks specific numbers. Please add quantitative support here.
Response:
Thank you for this valuable suggestion. We have revised the manuscript to include semiquantitative data supporting the claim. Specifically, evidence indicates that individuals residing above 3,500 m may have a resting minute ventilation nearly twice that of individuals at sea level, resulting in a significantly higher daily inhaled air volume. This is now supported by the following reference: West JB. Oxygen enrichment of room air to relieve the hypoxia of high altitude. Respir Physiol. 1995;99(2):225–232.
We also expand upon this physiological phenomenon in the discussion section, linking it to the context of pollutant exposure in high-altitude settings.

Comment 4. In line 95, the authors state: “It is well known that these countries often enforce less stringent regulations, frequently exceeding both AQS and WHO guidelines.” Please clarify that countries are exceeding their own AQS.
Response:
Thank you for this clarification. We have revised the sentence to specify that air pollution levels in these countries often exceed their own national air quality standards (AQS), when such standards exist, as well as the WHO guidelines. This change improves both the precision and interpretability of the statement.

Comment 5. In the beginning of the introduction, air pollution is discussed only in terms of mortality. Please mention other associated health outcomes beyond mortality and cardiorespiratory diseases.
Response:
Thank you for this insightful comment. We have revised the introduction to emphasize that air pollution’s health effects extend beyond mortality and classical cardiorespiratory outcomes. The updated text now highlights additional impacts on metabolic, reproductive, and neurodevelopmental health, supported by the following references you recommended:

  • 1016/j.envres.2021.111487
  • 1016/j.apr.2024.102377

Comment 6. The review mentions only certain pollutant groups, yet one included study (Goddard & Goddard, 1979) analyzes H₂S. Please revise the eligibility criteria or justify its inclusion.
Response:
Thank you for this helpful observation. Our review primarily focused on the six criteria pollutants due to their public health relevance. Although the study by Goddard & Goddard (1979) focuses on hydrogen sulfide (H₂S), it presents a generalized approach for converting pollutant concentrations into molecular units—a method applicable to any gas when molecular weight and ambient concentration are known. We have now clarified this in Table 1.

Comment 7. The methodology section does not mention any quality appraisal of the included studies.
Response:
Thank you for this relevant observation. In line with the JBI methodological framework for scoping reviews, no formal quality assessment of the included studies was performed. The primary objective was to map and synthesize available evidence, not to appraise study quality or risk of bias. This has now been clearly stated in the methodology section.

Comment 8. Lines 161, 169, 193, among others, show “Error! Reference source not found.” Please revise. Also revise the formatting of tables, especially Table 2.
Response:
Thank you for pointing this out. We have corrected all formatting issues related to broken internal cross-references (e.g., “Error! Reference source not found.”). Additionally, we revised the formatting of all tables to ensure proper alignment, consistent units, and clear column headings. In particular, Table 2 has been updated to reflect that it includes multiple pollutants—not just particulate matter. The caption now reads accordingly.

Comment 9. In Table 1, the phrase “rationale of proposed” is poorly worded.
Response:
Thank you. The wording has been corrected to "Basis for Model Adjustment" for clarity and improved readability.

Comment 10. The paragraph between lines 219–228 is repeated.
Response:
Thank you for noticing this duplication. The repeated paragraph has been removed from the revised manuscript.

Comment 11. In line 299, you state: “The WHO AQS (2000) guidelines acknowledge altitude...” What about the 2021 guidelines?
Response:
Thank you for pointing this out. We have revised the manuscript to clarify that only the 2000 WHO air quality guidelines explicitly consider altitude as a factor influencing air pollutant concentrations. The updated 2021 guidelines do not mention altitude in this context. This distinction is now clearly stated in the text.

Comment 12. In lines 315–318, the description of Figure 1 lacks detail.
Response:
Thank you for this helpful suggestion. We have expanded the description of Figure 1 in the main text (Section 3.2, lines 317–327), including more detailed information about population size and geographic distribution of people living at ≥3,500 m, disaggregated by country where possible.

Comment 13. In lines 348–349, the statement about China’s AQS should be more quantitative.
Response:
Thank you for this important point. We have revised the manuscript to include a quantitative comparison:
“China’s annual PM₂.₅ standard (35 µg/m³) is seven times more permissive than the WHO 2021 guideline (5 µg/m³).”
This provides a clearer sense of scale in regulatory stringency.

Comment 14. In lines 382–383, you mention Peru’s AQS and refer to WHO Interim Target I (2005). Why not compare with the updated 2021 guidelines?
Response:
Thank you for this observation. We referred to the 2005 WHO guidelines because Peru’s AQS, established in 2017, were based on those earlier targets. However, it is relevant to note that WHO Interim Target 1 values remained unchanged in the 2021 update. We have clarified this in the text to prevent confusion and ensure proper historical context.

Comment 15. In line 387–388, you mention “the WHO recommendation by a factor of ten.” Which recommendation? Clarify whether this refers to the 2005 or 2021 guidelines.
Response:
Thank you for pointing this out. We have clarified in the revised manuscript that the “factor of ten” comparison refers specifically to the current 2021 WHO guideline for PM₂.₅ (annual mean of 5 µg/m³). The sentence has been revised to explicitly mention the 2021 guidelines to avoid ambiguity.

Comment 16. A major issue that requires attention is the high redundancy between the Results and Discussion sections. Much of the discussion simply repeats content from the results, particularly descriptions of individual studies. I recommend merging these into a single section to streamline the content and reduce repetition. In this new combined section, I also recommend adding a subsection with recommendations. These recommendations would benefit from the authors’ personal perspectives. Based on your findings, do you suggest that AQS should vary by altitude tiers (e.g., 2,000–3,500 m, >3,500 m), or should they be recalculated for each city specifically? How could such recommendations be implemented in countries with highly diverse topography? Offering your perspective here would strengthen the policy relevance of your study. Consider this, as well as any other recommendations relevant to the topic.

Response: 

We sincerely thank the reviewer for this insightful observation. In response to this comment—as well as similar concerns raised by the other three reviewers—we have substantially reorganized the Discussion section to enhance clarity, coherence, and relevance. The revised version is now structured into four focused subsections: (1) Altitude-Specific AQS Adjustment Proposals, (2) Regulatory Frameworks, (3) Implications and Research Needs, and (4) Strengths and Limitations. This new structure facilitates a clearer presentation of the key points, reduces redundancy, and incorporates the authors’ perspectives and recommendations, as suggested.

Comment 17. In lines 432–434, the authors write: “Furthermore, these challenges may be exacerbated by climate change and urban expansion, as HA environments could become increasingly sought after as refuges from extreme heatwaves.” I believe it is important not only to mention this climate migration dynamic, but also to discuss how climate change-related meteorological shifts can influence local air pollution levels. Changes in weather patterns can affect the formation, dispersion and deposition of pollutants and worsen air quality, and this behavior is region-specific (relevant references: 10.1080/09603123.2025.2486598; 10.3390/atmos16040363).

Response

Thank you for this valuable suggestion. We have revised the corresponding paragraph to incorporate the broader implications of climate change on air quality in high-altitude environments. Specifically, we now address how projected meteorological changes—such as in temperature, wind patterns, and humidity—can significantly affect the formation, dispersion, and deposition of air pollutants. We also highlight the region-specific nature of these effects and reference recent studies demonstrating the importance of localized analyses. The revised paragraph also expands on climate-related migration dynamics in the context of glacial retreat, water scarcity, and shifting economic pressures. Both references you provided have been cited in the updated version (lines 447 - 458).

Reviewer 3 Report

Comments and Suggestions for Authors

Dear Authors, 

This systematic review addresses the crucial issue of modifying air quality standards (AQS) for high-altitude areas, where atmospheric factors and human susceptibility differ markedly from those at sea level. The topic is timely and significant, and the review is prepared well overall.

However, I suggest the authors consider and respond to several questions to enhance the robustness of the paper. Moreover, including relevant studies from the South Asian region-known for its high vulnerability to air pollution-in the introduction would enrich the context and add valuable perspective to the discussion.

I have attached my comments.

Thank you.

Author Response

We sincerely thank the revision and the editorial team for their time, thoughtful evaluation, and constructive feedback on our manuscript. We truly appreciate the opportunity to improve our work based on your valuable insights. We have carefully addressed all comments raised by the reviewers. As part of our resubmission, we are providing the new versions of the revised manuscript (ijerph-3649864 V2). 

Below, we provide a detailed, point-by-point response to each comment, indicating how and where the manuscript has been revised accordingly.

Comment 1:
This review highlights atmospheric conditions (for example, pressure, photochemical activity) and physiological adaptations (for example, increased ventilation) at high altitudes. How do you propose integrating these two domains (physical environment and human physiology) into a unified framework for adjusting AQS? Can you add a paragraph relating to this?

Response:
Thank you for this valuable comment. We agree that integrating atmospheric conditions and physiological responses is essential to developing a more comprehensive framework for altitude-specific AQS. however, is a complex challenge. In response, we have added a new paragraph in the “Implications and Research Needs” subsection, proposing a conceptual framework (602-609) incorporating health outcomes and physiological adaptations offers a more relevant and comprehensive basis.

Comment 2:
The section on respiratory physiology (lines 71–77) mentions native high-altitude populations and their adaptations. If possible, can you include any studies distinguishing native and non-native HA populations regarding pollutant vulnerability or regulatory needs?

Response:
Thank you for this insightful suggestion. We have expanded the discussion in the section on respiratory physiology to include relevant studies that differentiate the vulnerability of native versus non-native high-altitude populations. Evidence suggests that long-term residents or high-altitude natives exhibit adaptive traits (e.g., increased lung volumes, ventilatory), which may confer differential susceptibility to pollutant exposure. In contrast, newcomers or lowland migrants may be more acutely affected. These considerations are now reflected in the revised text (lines 506-514), along with relevant citations.

Comment 3:
The manuscript highlights the lack of high-altitude (HA)-specific air quality regulations, particularly in countries within the South Asian region. Given studies from South Asia that investigate pollutant behavior at different altitudes. Several studies have examined variations in specific air pollutants, such as ..., between HA and sea-level locations. For instance, Shelton et al. (2022), in their study titled "Seasonal variability of air pollutants and their relationship to meteorological parameters in an urban environment," compared pollutant levels in two urban cities—one at sea level and one at high altitude—and analyzed how meteorological conditions influenced pollutant concentrations. Incorporating such studies would strengthen the geographic representation and practical relevance of your review.

Response:
Thank you for this valuable suggestion. We have incorporated broad implications of climate change on air quality in high-altitude environments. Specifically, we now address how projected meteorological changes, such as in temperature, wind patterns, and humidity, can significantly affect the formation, dispersion, and deposition of air pollutants. We also highlight the region-specific nature of these effects and reference recent studies demonstrating the importance of localized analyses. The revised paragraph also expands on climate-related migration dynamics in the context of glacial retreat, water scarcity, and shifting economic pressures. Both references you provided have been cited in the updated version (lines 447 - 458).

Comment 4:
I also suggest including some studies from a representative Australian region. There are a few studies related to air pollution in the Blue Mountains, Australia HA region (600–1100 m a.s.l). For example, Liu et al., ... -cost sensors to high PM2.5 concentrations during bushfire and haze events, a comparison of PM2.5 between Chinese cities and Blue Mountain regions.

Response:
Thank you for this excellent suggestion. This case highlights how topography and fire-related events influence pollutant levels even at moderate elevations. Its inclusion broadens the geographical and contextual scope of the review and illustrates how local environmental factors interact with altitude-related pollution patterns.

Comment 5:
Given the physiological adaptations of HA populations, how do you recommend integrating epidemiological and biological data into air quality thresholds? Is there a preferred approach identified?

Response:
Thank you for this important question. We have added a paragraph in the “Implications and Research Needs” subsection to address this point. While there is currently no standardized method for integrating physiological or epidemiological data into AQS development, we propose the use of health outcomes that reinforce adjustment factors derived from high-altitude studies. These could include, pulmonary function metrics, and chronic disease prevalence data to inform exposure-response relationships. We emphasize the need for future research to develop such integrated models.

Comment 6:
Based on your review, how do you suggest keeping international air quality guidelines consistent while also making changes for high-altitude areas? Could making special rules for different places cause confusion or problems with regulations?

Response:
Thank you for raising this critical regulatory consideration. We have expanded the discussion on this issue under “Altitude-Specific AQS Adjustment Proposals.” We acknowledge that introducing localized or altitude-specific AQS presents challenges to regulatory harmonization. However, we argue that international guidelines—such as those from WHO—could include optional altitude-based adjustment coefficients or technical annexes tailored for high-altitude settings, similar to how occupational exposure limits are adjusted for different physical workloads. This would allow flexibility without undermining overall consistency.

Comment 7:
Please check the unit “μg/m³” throughout the manuscript. There appear to be some typos or inconsistencies in how it is written in different sections.

Response:
Thank you for pointing this out. We have carefully reviewed the entire manuscript to ensure that all units are written consistently and correctly throughout the text, tables, and figures.

Reviewer 4 Report

Comments and Suggestions for Authors

The authors address an important but often overlooked topic by reviewing models proposed for adjusting air quality standards (AQS) in high-altitude environments. This is important because high-altitude atmospheric conditions can affect in how air pollutants form and move. However, the authors identified only six relevant articles for review, which shows this topic needs more attention. They also compare AQS for different pollutants in three high-altitude countries with WHO guidelines.

Overall, I recommend this article for publication. The topic is relevant to IJERPH’s scope and likely of interest to readers. The paper is well-organized, focused, and mostly well-written. I only suggest a few minor edits:

  • Some references are missing and show as "Error! Reference source not found" (e.g., lines 162, 169, and 293).
  • Some abbreviations are introduced multiple times unnecessarily, such as "air quality standards (AQS)" on lines 44, 112, 348, and 416, on the other hand "particulate matter (PM)" on line 304 is introduced, although the term is used earlier without the abbreviation.
  • Provide references for the equations mentioned on lines 255 and 256.
  • Make sure consistency in unit formatting throughout the manuscript. For example, line 364 and 367 use " 𝜇𝑔 𝑚⁻3 ", but Table 2 includes both "μg/m³" and "𝒖𝒈/𝒎𝟑". Choose one format and apply it consistently.

Author Response

We would like to sincerely thank the reviewer for their thoughtful and encouraging feedback on our manuscript. We greatly appreciate your recognition of the relevance and organization of our work, as well as your constructive comments aimed at improving the clarity and consistency of the paper.

Although your suggestions were minor, they were highly valuable in refining key details, and we have carefully addressed each point. We believe these revisions have enhanced the overall quality and presentation of the manuscript. Our detailed responses to your comments are provided below. 

Comment 1. Some references are missing and show as "Error! Reference source not found" (e.g., lines 162, 169, and 293).

Response:
Thank you for pointing this out. These issues were due to broken internal cross-references during the document formatting process. We have carefully reviewed and corrected all such errors throughout the manuscript to ensure that every reference is properly displayed and linked.

Comment 2:
Some abbreviations are introduced multiple times unnecessarily, such as "air quality standards (AQS)" on lines 44, 112, 348, and 416. On the other hand, "particulate matter (PM)" is introduced on line 304, although the term is used earlier without the abbreviation.

Response:
We appreciate this observation. We have revised the manuscript to ensure that abbreviations are defined only at their first appearance and are used consistently thereafter. Redundant definitions have been removed, and missing ones have been corrected to improve clarity and flow.

Comment 3:
Provide references for the equations mentioned on lines 255 and 256.

Response:
Thank you for this helpful suggestion. We have added appropriate references to support the equations presented on lines 255 and 256. These references clarify the source and applicability of the conversion formulas used for pollutant concentrations.

Comment 4:
Make sure consistency in unit formatting throughout the manuscript. For example, line 364 and 367 use " ?? ?⁻3 ", but Table 2 includes both "μg/m³" and "??/??". Choose one format and apply it consistently.

Response:
Thank you for your attention to detail. We have reviewed the manuscript thoroughly and standardized the formatting of all units. The notation "μg·m−3" has been adopted consistently throughout the text, tables, and figures to ensure uniformity and readability.

Round 2

Reviewer 1 Report

Comments and Suggestions for Authors

After the authors have taken the reviewers' comments from the previous report into consideration in the latest revision, the quality of this manuscript has improved significantly.

Author Response

We sincerely thank you for the positive and encouraging feedback. We are pleased to hear that the quality of the manuscript has improved significantly after addressing the previous comments. We greatly appreciate the time and effort the reviewer has devoted to evaluating our work and providing constructive suggestions, which have contributed to enhancing the overall quality of the manuscript.

Reviewer 2 Report

Comments and Suggestions for Authors

The manuscript has been substantially improved, and the overall quality of the text is now much better. The authors have addressed the previous comments appropriately, and the content is clearer and more cohesive. However, it is important to note that Figure 1 currently disrupts the flow of the text. In the submitted file, the figure appears in the middle of a paragraph, breaking the continuity of the discussion. The authors are encouraged to carefully check the formatting when saving the final version as a PDF to ensure that layout issues are corrected prior to submission. Aside from this formatting issue, in my point of view the manuscript doesn't require any other correction.

Author Response

Thank you very much for your kind and constructive feedback. We appreciate your positive evaluation of the revised manuscript.

As suggested, we have carefully reviewed and corrected the formatting issue related to figure 1. The figure is now placed in closer alignment with the corresponding description in the text and no longer interrupts the paragraph. This adjustment improves the flow and readability of the discussion.

Additionally, we ensured that the formatting in both the Word file and the final PDF version is consistent, with no layout distortion or displacement of figures. The overall structure has been reviewed to guarantee a smooth reading experience.

We are grateful for your recommendation, which helped us enhance the presentation quality of the manuscript.

Reviewer 3 Report

Comments and Suggestions for Authors

I am satisfied with the authors’ responses. However, I noticed that some of the claims made in their replies are not properly cited in the revised manuscript, and there are inconsistencies between their responses and the updated reference list. I recommend that the authors carefully revise the manuscript to ensure that all referenced claims are properly cited, and that the reference list is updated accordingly to reflect these sources.

Author Response

We sincerely thank the reviewer for their continued time and valuable feedback during this second round of revision. We appreciate your observation regarding the mismatch between some claims in our previous responses and the corresponding citations in the revised manuscript, as well as the absence of some references in the reference list.

In response, we have thoroughly revised the manuscript to ensure that:

  • All claims made in the revised version are now properly supported with corresponding in-text citations.
  • We have emphasized the importance of seasonal variability in influencing air pollution patterns, as demonstrated by Shelton et al. in Sri Lanka, a country characterized by variable topography and altitude. This reference has been included in the Discussion section (line 450, reference 93).
  • Likewise, the study by Liu et al. on the use of low-cost sensors offers a relevant approach for assessing pollution in high-altitude settings. This reference has been incorporated into the Research Needs section (line 619, reference 131).
  • The reference list has been carefully reviewed and updated to ensure that all cited sources are accurately and consistently reflected.

We have uploaded a revised version of the manuscript with tracked changes to clearly indicate all modifications made in response to your comment.

Thank you again for helping us improve the clarity and scientific rigor of our work.
